# Deep Convolutional Neural Network for HEp-2 Fluorescence Intensity Classification

**Donato Cascio \***, **Vincenzo Taormina and Giuseppe Raso**

Department of Physics and Chemistry, University of Palermo, 90128 Palermo, Italy;
taormina.maltese@gmail.com (V.T.); giuseppe.raso@unipa.it (G.R.)
**\*** Correspondence: donato.cascio@unipa.it



**Featured Application: In this paper we describe an automatic system for fluorescence intensity classification to support the autoimmune diagnostics in HEp-2 image analysis. The system is based on the use of a pre-trained convolutional neural network (CNN) to extract features and a support vector machine (SVM) classifier for the positive or negative association.**

**Abstract:** Indirect ImmunoFluorescence (IIF) assays are recommended as the gold standard method for detection of antinuclear antibodies (ANAs), which are of considerable importance in the diagnosis of autoimmune diseases. Fluorescence intensity analysis is very often complex, and depending on the capabilities of the operator, the association with incorrect classes is statistically easy. In this paper, we present a Convolutional Neural Network (CNN) system to classify positive/negative fluorescence intensity of HEp-2 IIF images, which is important for autoimmune diseases diagnosis. The method uses the best known pre-trained CNNs to extract features and a support vector machine (SVM) classifier for the final association to the positive or negative classes. This system has been developed and the classifier was trained on a database implemented by the AIDA (AutoImmunité, Diagnostic Assisté par ordinateur) project. The method proposed here has been tested on a public part of the same database, consisting of 2080 IIF images. The performance analysis showed an accuracy of fluorescent intensity around 93%. The results have been evaluated by comparing them with some of the most representative state-of-the-art works, demonstrating the quality of the system in the intensity classification of HEp-2 images.

**Keywords:** IIF images; autoimmune diseases; Convolutional Neural Network (CNN); SVM; accuracy; receiver operating characteristic (ROC) curve

## 1. Introduction

Autoimmune diseases are several chronic disorders, and there are over 80 different types, some of which are disabling. Antinuclear antibodies are significant biomarkers in the diagnosis of autoimmune diseases in humans, which is performed by means of Indirect ImmunoFluorescence (IIF) test with human epithelial cells (HEp-2 cells) as antigens. The evaluation of antinuclear antibodies (ANAs) consists of the analysis of the fluorescence intensity and the staining patterns. IIF is a test having high sensitivity, but only analytical and not diagnostic specificity, since the positivity for ANA does not automatically confirm the presence of autoimmune disease; indeed, the ANA may be present even in healthy subjects [1,2]. In the past years, a great deal of effort was put into research regarding Indirect Immunofluorescence techniques, with the aim of development of computer-assisted diagnosis (CAD) systems [3].

In the clinical practice, IIF samples are categorized into a specific number of levels based on the visual assessment of their fluorescent intensity compared to a set of negative and positive controls.

As it is aimed at identifying the patient's positivity or negativity to the test, the fluorescence intensity classification phase is very important. Moreover, as regards the CAD system, it will be the result of this phase to establish (in the case of positive output) if the execution of the analysis steps aimed at identifying the staining patterns present in the image will be carried out. Figure 1 shows examples of each class.

In the literature, there are few scientific works on the automatic analysis of fluorescence intensity in IIF images, and to date, to our knowledge, no article has been published with reference to a public database.

Di Cataldo et al. [4] presented a method, ANAlyte, which is able to characterize IIF images in terms of fluorescent intensity level and fluorescent pattern without any user-interactions. They obtained overall accuracy of fluorescent intensity around 85%.

Elgaaied Benammar et al. [5] have optimized and tested a CAD system on HEp-2 images, which is able to classify the fluorescence intensity. The system classifies positive and negative images using one support vector machine (SVM) classifier. Results showed 85.5% accuracy in intensity fluorescence detection.

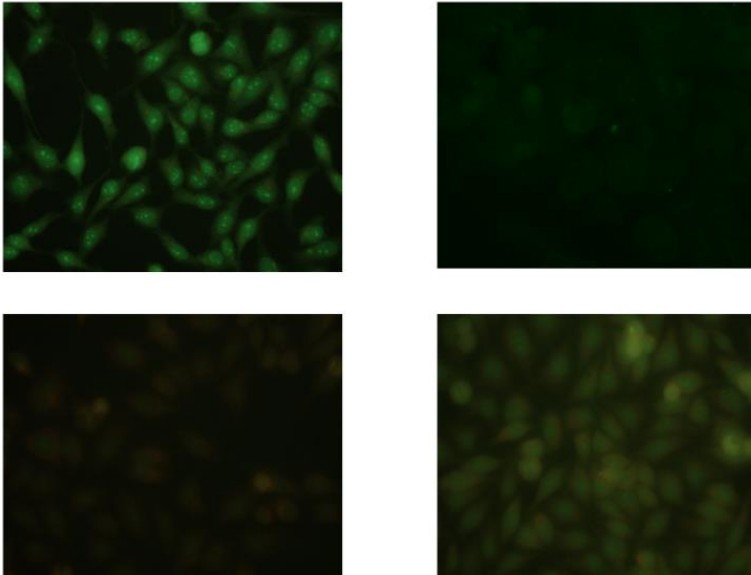

**Figure 1.** Indirect ImmunoFluorescence (IIF) images with different fluorescence intensity. At the top are two positive examples, below are two negative examples.

In one of our previous works [6], the analysis of fluorescence intensity has been addressed. The goal has been achieved by performing a preprocessing phase for the image, extracting a considerable number of features, and implementing an SVM classifier. To achieve a reduction in complexity and an appropriate selection of features, the linear discriminant analysis (LDA) method was used. The results obtained show an accuracy of 87% and an $A_Z$ area under the receiver operating characteristic (ROC) curve equal to 91.4%.

In recent scientific research on pattern recognition, Convolutional Neural Networks (CNNs) have been proved to be efficient and reliable models to achieve remarkable performance for image classification and object detection tasks [7]. Moreover, it has been demonstrated that pre-trained CNN architectures can play an important role in terms of features extractors, and allow high classification performance.

In order to address the fluorescence intensity classification, in this work, a method based on deep CNN is implemented. The distinctive appearance differences between image classes are represented through the learned features from pre-trained CNNs. The positive/negative image classification is carried out using a support vector machine classifier (SVM).

## 2. Materials and Methods

### 2.1. Database

The development of a classification system is intimately linked to the database used [8,9]. In this work, a public dataset (provided by AIDA: AutoImmunité, Diagnostic Assisté par ordinateur project) was used [5]. At the time, this dataset is the only public dataset with positive and negative wells, while the other two main HEp-2 image public datasets [10,11] contain only positive and weak positive images, but not negative cases. This database consists of two parts—one part is public, the other is private.

The AIDA HEp-2 public Database is a subset of the full AIDA database, where three physician experts have expressed, independently, unanimous opinion on reporting. This database is available to the scientific community, and to our knowledge, until today, it is the biggest HEp-2 image public database, with a total of 2080 images; 1498 images show positive fluorescence intensity, 582 show negative. These images correspond to the routine IIF technique performed in different hospitals for autoimmune diseases diagnosis, and were thus reported by senior immunologists. The database contains fluorescence positive sera with a variety of more than twenty staining patterns. They carried out serial dilutions and considered the dilution of 1/80 as positive. HEp-2 images have been acquired by means of a unit consisting of a fluorescence microscope (40-fold magnification). The images have 24 bits color depth and are stored in common image file formats.

The public database can be downloaded, after registration, from the download section of the site (http://www.aidaproject.net/downloads). The private part of the AIDA Database is structured in the same way as the public part, and it has about 20,000 images. However, among all these images, only a number of about 3000 have triple concordance of reports. This part of the AIDA database is only accessible to the partners who participated in the project.

The classification system here proposed has been trained using the private part of the database and tested on the public part.

### 2.2. Statistics

Whenever you want to discriminate within two classes, for example positive/negative, as in this case, the evaluation of the performance of a diagnostic system is generally expressed by a pair of indices—sensitivity and specificity. The sensitivity of a test is the fraction of recognized positive images (true positives) on the total number of positive images (true positives + false negatives), namely:

$$\text{Sensitivity} = \frac{\text{True Positives}}{\text{True Positives} + \text{False Negatives}} \tag{1}$$

The specificity of a test is the fraction of recognized negative images (true negatives) on the total number of negative images (true negatives + false positives), that is:

$$\text{Specificity} = \frac{\text{True Negatives}}{\text{True Negatives} + \text{False Positives}} \tag{2}$$

The need to obtain diagnostic systems with high sensitivity and high specificity leads to defining the accuracy that can be seen as the weighted sum of the two previous indices. Theaccuracy is defined as follows:

$$\text{Accuracy} = \frac{N_+ \, \text{Sensitivity} + N_- \, \text{Specificity}}{N_+ + N_-} \tag{3}$$

where $N_+$ represents the number of positive images and $N_-$ the number of negative images.

A diagnostic system generally produces an output value that must be compared to a threshold value in order to define the positivity or negativity of the image. This leads to variability of the performances according to the assigned threshold value.

An additional way to evaluate the performance of an automated system is represented by the Receiver Operating Characteristic (ROC). When the threshold value changes, different pairs of sensitivity and specificity will define the specific ROC. Another measure normally used to describe the performance of a system is therefore the area below the ROC curve, generally indicated with $A_Z$ [12].

### 2.3. Preprocessing

In this work, the processing was conducted using only the green channel, as it contains all the information present in the images; the other two channels essentially lead to noise contributions [13]. In order to obtain a robust system, and in particular less dependent on contrast variations (very present in IIF test images), in this work the image has been transformed using the following contrast stretching method:

$$T(x,y) = \frac{I(x,y) - \min[I(x,y)]}{\max[I(x,y)] - \min[I(x,y)]} \times 255 \tag{4}$$

where $I$ is the input image, $T$ is the image after the transformation, min and max represent, respectively, the minimum and maximum intensity of the input image. The normalization is carried out at the maximum value of 255, since the IIF always images have an 8-bit depth. Studies were initially conducted, in which noise reduction filters, such as the Gaussian filter and the median filter, were applied to the image. However, it was observed that in the use of pre-trained CNNs, filtering had a negative impact in terms of intensity classification performances.

### 2.4. Deep CNN

Deep learning, in particular Convolutional Neural Networks (CNN), is a validated image representation and classification technique for the analysis of biomedical images and applications. In recent years, the scientific community has produced many encouraging works that report new and more recent state-of-the-art performance on quite challenging problems in this domain [14–16]. The main reason behind this stream of work is probably because the effective task-dependent image features can be directly or intrinsically learned through the hierarchy of convolutional kernels inside CNN. Most deep learning methods use neural network architectures, which is why deep learning models are often referred to as deep neural networks. The term "deep" usually refers to the number of layers hidden in the neural network. Traditional neural networks contain only 1–2 hidden layers, while deep networks can contain up to 150. Deep learning models are trained using large labeled data sets and neural network architectures that learn features directly from data without having to manually extract them.

One of the most common types of neural networks is known as a convoluted neural network (CNN or ConvNet). A CNN conveys the characteristics learned with the input data and uses the convolutional layers in 2D, which make this architecture suitable for 2D data processing, such as images.

CNNs eliminate the need for manual feature extraction [17–19], so the user does not have to identify features used for image classification. In fact, it is possible to use the power of the pre-trained networks, without investing time and effort in training, to implement the extraction phase of the characteristics. Feature extraction can be the fastest way to use in-depth learning. The operation of CNN is based on the extraction of the features directly from the images. The automatic extraction of the features allows the high precision of the deep learning models intended for artificial vision activities, such as the classification of objects.

In this work, it was decided to perform fluorescence intensity classification by analyzing the whole image without applying a segmentation to the cells. As it is known that the problem of finding the best set of discriminating features for a given classification problem is very complex, we have decided, in line with recent scientific trends, to use pre-trained CNNs to extract features.

### 2.5. Pre-Trained Networks Used

In this work, several of the best-known pre-trained CNN architectures have been used in order to identify the most performing one.

Moreover, for each architecture used, the various layers were analyzed in order to find the most discriminating one for the classification problem addressed.

The families of CNN architectures that have been used are the following:

- **AlexNet** [20]: This network has been trained on 1.3 million high-resolution images in the LSVRC-2010 ImageNet training set into the 1000 different object classes. Rectified Linear Units (ReLU) is used as a non-linear activation function at each layer;
- **googleNET** [21]: This architectures makes use of so-called inception blocks. Inception blocks can be interpreted as a network-in-a-network, where the input is branched into several different convolutional sub-networks, which are concatenated at the end of the block;
- **VGG** [22]: The main idea of this architecture is to increase depth and reduce the dimension of convolution filters. The image is passed through a stack of convolutional (conv.) layers, where are usedfilters with a very small receptive field: $3 \times 3$ (which is the smallest size to capture the notion of left/right, up/down, center);
- **ResNet** [23]: This architecture is currently the best performing deep architecture, being the winner of the ImageNet challenge in 2015. The authors propose deeper CNN for solving the problem of performance degradation due to depth with a residual learning framework. Instead of hoping each few stacked layers directly fit a desired underlying mapping, the authors explicitly let these layers fit a residual mapping;
- **Densenet** [24]: This pre-trained CNN has an architecture that connects each layer with all the others with depth greater than its own (feed forward mode). One of the peculiarities is that the concatenation of the features from the previous layers is made by concatenating them;
- **Sequeezenet** [25]: The authors have built a smaller architecture with three main advantages: smaller CNNs require less communication across servers during distributed training; smaller CNNs require less bandwidth to export a new model from the cloud to an autonomous car; smaller CNNs are more feasible to deploy on FPGAs and other hardware with limited memory.

### 2.6. SVM Classification

The effectiveness of the characteristics extracted from the different CNNs has been used in order to associate the generic image with positive or negative classes. The main feature of the SVMs, which led them to immediate success, is the fact that they can achieve high performance in practical applications. Furthermore, their simplicity, in terms of parameters, makes it possible to tackle complex classification problems, in which there are—as in our case—a large number of input features. This need for simplicity has led us to implement a SVM classifier with linear kernel [26,27], the simplest in terms of parameters to search.

## 3. Results

The fluorescence intensity classification method, described in Section 2, was analyzed using all images (2080 images) in the public AIDA database. Performance analysis was differentiated for each pre-trained network used.

Table 1 shows the accuracy obtained for the various CNN used. Furthermore, the table shows the network layer numbers and the layer that has returned the most discriminating features (best layer).

The best configuration, obtained with the densenet201 CNN, showed a sensitivity in the recognition of positive images equal to 96.1%, while with regard to the ability to identify the negatives, this configuration showed a specificity of 84.4%. The ROC (Receiver Operating Characteristic) curve in the Figure 2 was obtained by plotting the true positive rate (TPR) against the false positive rate

(FPR) at various threshold settings. The area under the curve value obtained was $A_Z = 0.974 \pm 0.003$. The accuracy value obtained was Accuracy = 92.8%.

**Table 1.** Classification accuracy for the pre-trained convolutional neural networks (CNNs) analyzed.

| Pre-Trained CNN | Depth | n. Layers | Best Layer Results | Accuracy |
|---|---|---|---|---|
| alexnet | 8 | 25 | conv3 | 90.3% |
| googlenet | 22 | 144 | incep_3a-output | 90.1% |
| vgg16 | 16 | 41 | drop7 | 90.3% |
| vgg19 | 19 | 47 | drop7 | 90.5% |
| resnet18 | 18 | 72 | res5b_relu | 92.3% |
| resnet50 | 50 | 177 | avg_pool | 92.2% |
| resnet101 | 101 | 347 | res5c_relu | 92.2% |
| sequeezenet | 18 | 68 | drop9 | 89.2% |
| densenet201 | 201 | 709 | bn | 92.8% |

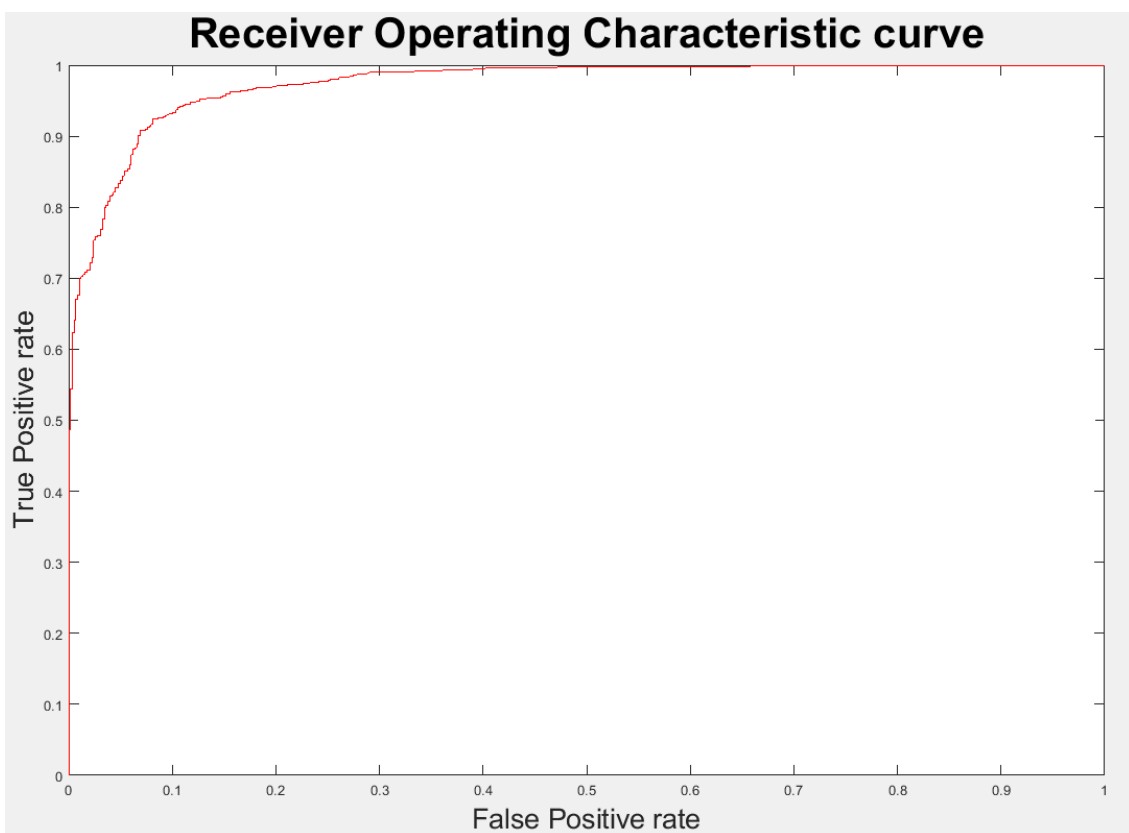

**Figure 2.** Receiver Operating Characteristic (ROC) curve of fluorescence intensity classification method.

Table 2 shows the results obtained and the performance comparison with other notable intensity fluorescence classification methods proposed in the literature in the last years. Due to the greater statistical weight of our result, Table 2 also shows that our method of fluorescence intensity classification performs better than the other methods analyzed.

From the comparison of the system proposed here with the other method [9] of using the same database, it is easy to deduce the quality of the analysis carried out and the potentialities of the method.



**Table 2.** Performance comparison with other methods.

|  | Images Dataset | Accuracy | A$_Z$ | Sensitivity | Specificity |
|---|---|---|---|---|---|
| Di Cataldo [4] | 71 | 85.7% | - | - | - |
| Benammar [5] | 1006 | 85.5% | - | 91.1% | 70.8% |
| Cascio [6] | 2080 | 87.0% | 91.4% | 92.9% | 70.5% |
| Our method | 2080 | 92.8% | 97.4% | 96.1% | 84.4% |

## 4. Discussion

In this paper the problem of automatic classification of fluorescence intensity in HEp-2 images, very important for the diagnosis of autoimmune diseases, has been addressed. To this end, CNN pre-trained networks were analyzed as feature pullers, combined with the traditional SVM (support vector machine) classifier. In particular, several more well-known pre-trained CNN architectures have been used. For each architecture used, the different layers were analyzed in order to find the most discriminating one for the classification problem addressed. In order to identify the set of features having the highest classification power for the characterization of the fluorescence intensity, the analysis carried out for the different configurations allowed the identification, in terms of network and layer, of the best-performing solution.

The classification system proposed here has been trained using the private part of the database and tested on the public part, to allow future comparisons and to avoid bias effects.

A comparison of the performances was presented with other recent state-of-the-art methods that highlight the quality of the proposed system and the very promising capabilities in discriminating the positive and negative images of the IIF test. In fact, the intensive analysis performed in this work, in terms of the pre-trained network number and the relative layers used as features extractors, has allowed us to obtain better fluorescence intensity classification performance than those obtained from other recent state-of-the-art methods. In order to have a further reference for the evaluation of the obtained performances, and a real perception on the complexity of the problem faced, we compare the performances of classification intensity obtained in the work of Benammar et al. [5] by two young immunologists, to those of researchers who were made to analyze images of the same AIDA database. The accuracy obtained by them for fluorescence intensity was 66% for both. Therefore, the results obtained demonstrate the effectiveness of the method presented here and the possibility that this can be used as a support tool in the diagnostic workflow of autoimmune diseases.

**Author Contributions:** D.C. conceived of the study, performed the statistical analysis, and drafted the manuscript. V.T. developed the software and optimized the parameters. G.R. participated in the design and coordination of the study, and has supported the writing of the manuscript.

**Funding:** This research received no external funding.

**Conflicts of Interest:** The authors declare no conflict of interest.

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
