# Peer review of "Deep Convolutional Neural Network for HEp-2 Fluorescence Intensity Classification"

_applsci, doi:10.3390/app9030408_

Round 1
Reviewer 1 Report
In this paper, the authors describe an automatic system for fluorescence intensity classification to support the autoimmune diagnostics in HEp-2 image analysis. The system is based on the use of a pre-trained CNN to extract features and an SVM classifier for the positive/negative association.
(1) Equation 1, sensibility or sensitivity?
(2) Equation 4, why times 255? What happened if the color depth is not 8.
(3) Section 2.5. Those pretrained net works are for ImageNet. Are they suitable for identifying to Fluorescence image?
(4) Section 2.6, why use SVM? Why not use fully-connected layer?
(5) Some related publications could be discussed, see “Image based fruit category classification by 13-layer deep convolutional neural network and data augmentation” and “Abnormal breast identification by nine-layer convolutional neural network with parametric rectified linear unit and rank-based stochastic pooling”
Author Response
We thank the reviewer for the attention he showed towards our manuscript and for the clarifications/suggestions proposed.
The following are the point-to-point answers
(1) Equation 1, sensibility or sensitivity?
Sensitivity. The error has been corrected. Thanks
(2) Equation 4, why times 255? What happened if the color depth is not 8.
The normalization is carried out at the maximum value of 255, since the IIF images have an always 8-bit depth.
Clarification added to the document.
(3) Section 2.5. Those pretrained net works are for ImageNet. Are they suitable for identifying to Fluorescence image?
The pre-trained networks used have been designed for ImageNet, but in this work they have been used as feature extractors; since they have been designed to identify numerous and different objects, CNNs, as feature extractors, are particularly effective. This is also demonstrated by the considerable use, in the most varied fields of research, of the networks pre-trained, in this way.
(4) Section 2.6, why use SVM? Why not use fully-connected layer?
CNNs used as extractors produce a large number of characteristics to be analyzed. Since in this work, the number of examples that could be used for supervised training was relatively small (almost 3000 images), this led to the choice to use a classifier with few parameters and therefore to the linear SVM classifier.
(5) Some related publications could be discussed, see “Image based fruit category classification by 13-layer deep convolutional neural network and data augmentation” and “Abnormal breast identification by nine-layer convolutional neural network with parametric rectified linear unit and rank-based stochastic pooling”
Suggested works have been inserted and discussed.
Reviewer 2 Report
The this Applied Sciences paper, Cascio et al, reported an automatic system as
Convolutional Neural Network (CNN) system to provide a clear cutoff of the
HEp-2 IIF images-based assay and further warranty a more accurate way to advance certain sort of autoimmune diseases diagnosis. The data is substantial and the manuscript was well written.
Some points:
1. It would be more convincing if the authors can provide a table which shows IIF sensitivity and specificity in a patients diagnostic setting, either from previous study or from the cohort from authors’ institute.
2. How about the sensitivity of CNN over other three methods in table 2?
3. Line 20 ‘trained” need to specify; Line 67 ‘are’ might change to ‘is’; Line 159 should have space at beginning; Line 170 should be space.
Author Response
We thank the reviewer for the attention he showed towards our manuscript and for the clarifications/suggestions proposed.
The following are the point-to-point answers
Some points:
1. It would be more convincing if the authors can provide a table which shows IIF sensitivity and specificity in a patients diagnostic setting, either from previous study or from the cohort from authors’ institute.
In order to have, a further reference for the evaluation of the obtained performances, and a real perception on the complexity of the problem faced, we report the performances of classification intensity obtained, in the work of Benammar et al [5], by two young immunologists to those who were made to analyze images of the same A.I.D.A. database. The accuracy obtained by them for fluorescence intensity was 66% for both. Therefore, the results obtained demonstrate the effectiveness of the method presented here and the possibility that this can be used as a support tool in the diagnostic workflow of the autoimmune diseases.
Explanation inserted in the manuscript.
2. How about the sensitivity of CNN over other three methods in table 2?
As required, the sensitivity and specificity values of the various methods have been included in table 2
3. Line 20 ‘trained” need to specify; Line 67 ‘are’ might change to ‘is’; Line 159 should have space at beginning; Line 170 should be space.
The errors reported have been corrected.
Round 2
Reviewer 1 Report
Accept